# Alpha-Synuclein Effects on Mitochondrial Quality Control in Parkinson’s Disease

**DOI:** 10.3390/biom14121649

**Published:** 2024-12-22

**Authors:** Lydia Shen, Ulf Dettmer

**Affiliations:** 1College of Arts & Sciences, Cornell University, Ithaca, NY 14853, USA; 2Ann Romney Center for Neurologic Diseases, Brigham and Women’s Hospital and Harvard Medical School, Boston, MA 02115, USA; udettmer@bwh.harvard.edu

**Keywords:** α-synuclein, Parkinson’s disease, mitochondrial dysfunction, PINK1/Parkin, PGC-1α, mitochondrial fragmentation, mitophagy

## Abstract

The maintenance of healthy mitochondria is essential for neuronal survival and relies upon mitochondrial quality control pathways involved in mitochondrial biogenesis, mitochondrial dynamics, and mitochondrial autophagy (mitophagy). Mitochondrial dysfunction is critically implicated in Parkinson’s disease (PD), a brain disorder characterized by the progressive loss of dopaminergic neurons in the substantia nigra. Consequently, impaired mitochondrial quality control may play a key role in PD pathology. This is affirmed by work indicating that genes such as PRKN and PINK1, which participate in multiple mitochondrial processes, harbor PD-associated mutations. Furthermore, mitochondrial complex-I-inhibiting toxins like MPTP and rotenone are known to cause Parkinson-like symptoms. At the heart of PD is alpha-synuclein (αS), a small synaptic protein that misfolds and aggregates to form the disease’s hallmark Lewy bodies. The specific mechanisms through which aggregated αS exerts its neurotoxicity are still unknown; however, given the vital role of both αS and mitochondria to PD, an understanding of how αS influences mitochondrial maintenance may be essential to elucidating PD pathogenesis and discovering future therapeutic targets. Here, the current knowledge of the relationship between αS and mitochondrial quality control pathways in PD is reviewed, highlighting recent findings regarding αS effects on mitochondrial biogenesis, dynamics, and autophagy.

## 1. Introduction

### 1.1. Parkinson’s Disease

Parkinson’s disease (PD) is the second most common neurodegenerative disorder after Alzheimer’s disease, projected to affect over 9.3 million individuals over age 50 by the year 2030 [1]. Neuropathologically, the disease is characterized by a progressive loss of dopaminergic neurons (DA neurons) in the midbrain substantia nigra pars compacta (SNpc) and the presence of cytoplasmic protein aggregates called Lewy bodies (LBs) in the remaining dopaminergic neurons, as well as other neuron types. Clinical motor symptoms of PD include resting tremor, rigor, bradykinesia, and postural instability [2]. PD is also associated with a number of nonmotor symptoms such as hyposmia, constipation, and rapid eye movement sleep behavior disorder, which can precede motor symptom onset by years or even decades [3]. While the hallmark motor symptoms of PD are mostly attributed to the loss of DA neurons, many of the nonmotor symptoms cannot be so easily explained. In fact, the etiology of PD has yet to be well defined. LBs are predominantly composed of misfolded alpha-synuclein (αS), implicating it as the key protein in PD [4]. However, the exact neurotoxic pathways through which αS causes neurodegeneration are not understood well [5]. There is increasing evidence that the interplay between αS misfolding and impaired mitochondrial biology may be at the center of PD pathology. This review article will provide an overview of our current knowledge about those underlying mechanisms—specifically, how αS may impact mitochondria maintenance, with an emphasis on studies published between January 2020 and November 2024—and identify areas that are not understood well.

### 1.2. Mitochondrial Impairment in PD

Mitochondria are dynamic organelles that play a critical role in many physiological processes within the cell, including energy metabolism, programmed cell death, cellular signaling, and calcium homeostasis [6]. Consequently, mitochondrial dysfunction is severely detrimental to cellular health, negatively impacting key biosynthetic processes and impairing overall homeostasis. Mitochondrial biology is intimately linked to reactive oxygen species (ROS), byproducts of normal aerobic metabolism that, if left unchecked, can cause irreversible oxidative damage to proteins and DNA [7,8,9]. Antioxidant defense mechanisms within the cell typically maintain low levels of ROS; however, in diseased states, this equilibrium is disrupted, leading to oxidative stress, and potentially contributing to the onset of neuronal disorders [10].

Accumulated evidence suggests that mitochondrial dysfunction may play a key role in PD pathogenesis. It is well established that exposure to toxins that inhibit complex I in the mitochondrial electron transport chain can induce dopaminergic cell loss and parkinsonism. For instance, exposure to 1-methyl-4-phenyl-1, 2, 5, 6-tetrahydropyridine (MPTP) induces PD-like symptoms and other pathological hallmarks of PD in humans and various animals [11,12,13,14,15]. Other complex I inhibitors, including rotenone [16], have also been shown to trigger nigrostriatal damage, and to a certain extent cause motor and nonmotor behavioral phenotypes related to PD [17]. In addition, genetic evidence highlights mitochondrial dysfunction as a contributor to and risk factor for PD. Genetic variation is estimated to explain 25% of the risk of developing PD, with approximately 15–25% of patients reporting a family history of the disease [18,19]. Of the gene mutations related to familial forms of PD, a substantial number are involved in mitochondrial processes, including PRKN, PINK1, PARK7, and LRRK2, further affirming the relevance of mitochondrial dysfunction to PD etiology [18].

Given these epidemiological and genetic links, mitochondria have received increasing attention in the characterization of the elusive pathogenesis of PD. Mitochondrial quality control (MQC) pathways help to ensure a healthy mitochondrial biology, which occurs in three major stages: mitochondrial biogenesis, mitochondrial dynamics, and mitochondrial autophagy [6,20]. A detailed understanding of how αS, the central protein in PD, affects MQC may provide further insight into PD pathogenesis, potentially highlighting therapeutic targets.

### 1.3. Alpha-Synuclein: The Key Protein in PD

αS is a small, 140-amino-acid protein that consists of three regions: an N-terminal region responsible for its membrane-binding behavior, the central “non-amyloid component (NAC)” region, and a C-terminal acidic tail [21]. It has a dynamic structure that varies based on its location within the cell: within the cytosol, it can exist as an unfolded monomer [22], but when bound to phospholipid membranes, it folds into an amphipathic helix [23] that may further assemble to form metastable tetramers [24,25]. αS binding to membrane phospholipids is believed to be mediated by the conserved KTKEGV repeat motif present in its N-terminal region based on its overall arrangement of charged, polar, and non-polar amino acids [26,27]. αS has a strong affinity for highly curved membranes that exhibit lipid packing defects, most importantly synaptic vesicles [28,29,30]. This binding affinity, which seems to underlie its predominant localization to presynaptic terminals, contributes to the general view of αS as a synaptic protein [31]. However, the precise function and behavior of αS is still under investigation, with its function both at and beyond the synapse poorly understood [25,32].

αS is the primary protein present in LBs, the hallmark aggregates in PD, implicating it as the key protein responsible for PD pathogenesis. Alongside αS aggregation, mitochondrial dysfunction is also associated with PD, as discussed earlier, but the connection between both aspects is poorly understood. Notably, it is still currently unknown whether αS misfolding and aggregation are upstream of mitochondrial problems, downstream of mitochondrial problems, or both [33,34,35,36,37,38]. Nuclear magnetic resonance has revealed that non-aggregated αS can bind to the outer mitochondrial membrane [39]. Others suggested that (non-aggregated) αS, expressed in yeast and human cells, is constitutively imported into mitochondria [40]. This translocation, if biologically relevant, might be aided by “cryptic targeting signals” in the αS sequence that can be recognized by mitochondrial import receptors [41]. It has been proposed that VDAC proteins are key mediators of αS translocation into mitochondria [42,43,44,45]. Frequently suggested downstream consequences of αS presence at/inside mitochondria include negative effects on complex I activity [46,47,48], and the impairment of mitochondrial respiration in general [43]. Interestingly, however, recent work is consistent with the presence of advanced αS aggregates, such as Lewy bodies, being associated with increased complex I expression—leading to the speculation that Lewy bodies may encapsulate damaged mitochondria and/or αS oligomers [49]. In addition to effects on respiration, an interaction of αS with mitochondrial DNA may occur [50].

Concerning the folding state of αS, pathogenic αS aggregates have been suggested to preferentially bind to mitochondria, reducing ATP production and inducing mitochondrial fragmentation [51]. A recent study demonstrated that the extracellular exposure of neurons to αS aggregates decreased mitochondrial gene expression, reduced the number of mitochondria, increased oxidant stress, and profoundly disrupted mitochondrial ATP production [52]. Other recent work emphasizes that αS aggregation, but not overexpression alone, induces mitochondrial degradation [53]. Interestingly, it was suggested that intracellular αS aggregation events may occur preferentially at mitochondrial membranes, aided by the lipid cardiolipin that is enriched at the inner membranes [54,55].

However, it is also well known that mutations in mitochondrial genes can cause PD (principally associated with αS aggregation), suggesting that mitochondrial dysfunction may precede αS dyshomeostasis [56]. Moreover, mitochondrial toxins can cause the downstream aggregation of αS [57]. Regardless, αS and MQC may affect each other, ultimately resulting in ROS and oxidative stress that may at least in part be responsible for DA neuron death. This review mainly focuses on how αS may affect MQC in biogenesis, dynamics, and autophagy.

## 2. Mitochondrial Biogenesis and PD

### 2.1. Mitochondrial Biogenesis

Mitochondrial biogenesis plays a critical role in mitochondrial quality control, as it is responsible for producing new, healthy mitochondria to replace damaged ones [58]. In PD, improper mitochondrial biogenesis may contribute to mitochondrial dysfunction, causing oxidative damage to DA neurons and subsequent neuronal death. Mitochondrial biogenesis can be activated by a number of factors, including environmental toxins, temperature and oxygen variation, nutrient availability, growth factors and hormones, and more [59]. As semiautonomous organelles, mitochondria require the expression of both the nuclear genome and the mitochondrial genome (mtDNA) to achieve their biological function. Mitochondrial biogenesis is therefore a complex process that involves the synthesis of mtDNA-encoded proteins, the synthesis of nuclear-DNA-encoded mitochondrial proteins, the synthesis of both the inner and outer mitochondrial membranes (IMM and OMM, respectively), and the replication of mtDNA. Mitochondrial biogenesis is regulated by the PGC-1α-NRF-TfamA pathway [60,61]. Adenosine monophosphate protein kinase (AMPK) and silent information regulator 1 (Sirt1) serve as upstream regulators of peroxisome-proliferator-activated gamma coactivator-1 alpha (PGC-1α), phosphorylating and deacetylating PGC-1α, respectively [62,63]. Upon activation, PGC-1α then stimulates nuclear respiratory factors 1 and 2 (NRF1 and NRF2), which in turn activate mitochondrial transcription factor A (Tfam) [64]. Tfam is a key transcription factor involved in mtDNA transcription and replication, as well as the transcription of nuclear-encoded mitochondrial proteins. Consequently, its activation induces the process of mitochondrial biogenesis [65]. PGC-1α is also known to induce the expression of genes involved in ion transport, mitochondrial protein translation, and protein import, as well as to stimulate respiratory function, making it a “master co-regulator of mitochondrial function” [66].

### 2.2. αS Effects on Mitochondrial Biogenesis in PD

Given the central role of proper PGC-1α regulation in mitochondrial biogenesis, many studies have centered around PGC-1α imbalances in PD [67]. Genome-wide meta-expression analyses of gene sets documented the downregulation of PGC-1α target genes in DA neurons of PD patients, congruent with the idea that altered PGC-1α expression contributes to PD pathogenesis [68,69]. PGC-1α knockout mice demonstrated greater sensitivity to oxidative stressors such as MPTP and kainic acid [70] but exhibited abnormal mitochondria in DA neurons [71]. Furthermore, the overexpression of PGC-1α was shown to protect neurons from oxidative stress-mediated cell death and to ameliorate αS-mediated toxicity by reducing αS oligomerization [70,72]. Put together, these findings suggest that dysfunctional PGC-1α and αS misfolding are both involved in PD pathogenesis. In fact, studies have proposed that nuclear αS may directly bind to the PGC-1α promoter sequence under oxidative stress, both in vitro and in vivo, reducing the expression of PGC-1α and its target genes, thereby impairing mitochondrial function (Figure 1a) [73]. The binding of αS to the PGC-1α promoter was also observed in brain tissues of PD patients [73]. αS may thus impair mitochondrial biogenesis by acting as a transcriptional repressor of PGC-1α, but further confirmation seems necessary to support this unexpected mechanism.

αS may also lead to abnormal mitochondrial biogenesis through interactions with the PINK1/Parkin signaling pathway. PINK1 and Parkin mutations were among the first to be linked with genetic PD, causing abnormal mitophagy, increased oxidative stress, and ultimately neuronal death [74,75]. PINK1 and Parkin have been suggested to regulate PGC-1α through the degradation of PARIS, a transcriptional repressor of PGC-1α [76,77]. PINK1 phosphorylates serine residues 322 and 613 of PARIS, allowing Parkin to ubiquitin-tag PARIS for proteasomal degradation [78,79]. The knockout of Parkin in mice may lead to a loss of DA neurons caused by the accumulation of PARIS and the resulting repression of PGC-1α [79]. Further, not only do PINK1 and Parkin promote mitochondrial biogenesis by inducing the degradation of PARIS, but Parkin may also directly interact with Tfam to enhance Tfam-mediated mitochondrial transcription [80,81]. It is widely accepted that PINK1 and Parkin have a significant neuroprotective effect, as loss-of-function mutations result in the creation of dysfunctional mitochondria. This is also supported by the fact that missense mutations in PINK1 and Parkin are some of the most common autosomal recessive forms of PD [82]. A study on neuronal cell models suggested that exposure to exogenous αS oligomers and fibrils reduces Parkin expression, inducing a decrease in PGC-1α levels (Figure 1a) [83]. This could suggest the existence of a vicious cycle in which αS causes Parkin downregulation, Parkin’s neuroprotective role is diminished, mitochondrial damage is further exacerbated, and αS misfolding and aggregation occurs by a yet-to-be-defined mechanism. Notably, however, the mechanism through which αS may downregulate Parkin is still unknown, though some studies have suggested that αS can cause post-translational modifications to Parkin [84]. To further complicate things, a recent study concluded that disease-causing multiplications result in Parkin accumulation in cells, highlighting the context-dependent effects of the disease [85]. Regardless, αS interactions with PGC-1α may contribute to mitochondrial dysfunction in PD, both through its role as a modulator of PGC-1α and via effects on Parkin in the PINK1/Parkin-PARIS-PGC-1α pathway. An interesting therapeutic idea relevant to this context involves a cell-permeable Parkin protein that, according to the authors, recovered damaged mitochondria by promoting mitochondrial biogenesis and mitophagy (Section 4) and thereby also suppressed αS accumulation in cells and animals [86]. Lastly, mitochondria rely on the continuous import of proteins for proper functioning. The translocase of the outer membrane 20 (TOM20) protein plays a pivotal role in initiating the import of precursor proteins into mitochondria. By binding to TOM20, αS has been suggested to interfere with proper protein translocation ([87]), which can be reversed by the overexpression of TOM20 ([88]).

## 3. Mitochondrial Dynamics and PD

### 3.1. Mitochondrial Dynamics

Within neurons, mitochondria are continually undergoing cycles of fusion and fission, collectively referred to as mitochondrial dynamics. Fusion refers to the combination of two mitochondria into a single mitochondrion, while fission refers to the separation of mitochondria into smaller parts [89]. A healthy balance between these two processes is crucial to the maintenance of normal mitochondrial function and homeostasis. For instance, fusion allows damaged mitochondria to regain functionality by exchanging functional proteins and non-damaged mtDNA with healthy neighboring mitochondria [90,91]. On the other hand, fission can isolate damaged mitochondria for degradation or protect them from further damage by redistributing damaged parts amongst other mitochondria. Imbalances in mitochondrial dynamics often lead to morphological alterations and mitochondrial dysfunction: impaired fusion causes mitochondrial fragmentation, whereas impaired fission leads to the formation of megamitochondria. Typical mitochondrial fusion is regulated by three proteins of the actin-related guanosine triphosphatase (GTPase) family: mitofusin 1 and 2 (MFN1 and MFN2) and optic atrophy 1 (OPA1) [92]. MFN1 and MFN2 participate in the fusion of the OMM, while OPA1 is involved in IMM fusion [93,94]. Meanwhile, the GTPase dynamin-related protein 1 (DRP1) is the main protein responsible for mitochondrial fission [95].

### 3.2. Mitochondrial Dynamics and αS in PD

Imbalances in mitochondrial dynamics are associated with PD phenotypes. For instance, an in vitro study with primary cortical neurons showed that exposure to rotenone, a complex I inhibitor associated with PD, or nitric oxide, a free radical, causes mitochondria to undergo rapid fission [96]. Meanwhile, the overexpression of MFN1 or the inactivation of DRP1 was able to prevent mitochondrial fragmentation and rescue neurons from degeneration [96]. Similarly, the administration of the complex I inhibitor MPP+ or the ROS-generating 6-OHDA to cultured neurons was associated with DRP1-dependent mitochondrial fragmentation [97,98]. Additionally, a variety of other studies have associated aberrant mitochondrial morphologies with PD, suggesting disruptions to normal fusion–fission dynamics (e.g., [99,100,101]). In neuroepithelial stem cells carrying a mutation to LRRK2, a gene implicated in PD pathogenesis, increased mitochondrial fragmentation was observed [101]. Furthermore, abnormally large mitochondria (size > 1 μm^2^) were observed in the soma of PGC-1α knockout mice, indicating impaired mitochondrial fission [71]. Aberrantly round mitochondria were also present in induced pluripotent stem cells with an A53T SNCA mutation or an SNCA locus triplication [102]. Thus, kinetic defects within mitochondria may play a crucial role in PD, leading to mitochondrial dysfunction and overall cellular dysfunction.

αS is suggested to interact with mitochondrial dynamics in several ways. The oligomerization of αS is associated with inhibited fusion and mitochondrial fragmentation, though the precise mechanisms remain to be established [103]. The overexpression of pathogenic αS, specifically A53T and A30P, was suggested to induce mitochondrial fragmentation by increasing the cleavage of OPA1, thus degrading it and preventing mitochondrial fusion from occurring (Figure 1b) [104]. However, the mechanism through which αS binds to or causes OPA1 cleavage is not currently known [105]. In abnormal mitochondrial fission, the role of αS is mostly speculative: it may either be independent of DRP1, dependent on DRP1, or some combination of both arrangements (Figure 1b) [106]. For instance, it was shown that αS can cause mitochondrial fragmentation in the absence of DRP1, directly interacting with mitochondrial membranes to induce fragmentation [107]. On the other hand, the overexpression of SNCA has been shown to increase the translocation of DRP1 to mitochondria, resulting in pro-fission effects [108]. Also, under normal conditions, the PINK1/Parkin signaling pathway promotes DRP1-dependent mitochondrial fission, so αS interference with PINK1/Parkin may influence fission [109]. Regardless, αS interactions with mitochondrial dynamic proteins are not yet fully understood and future work is necessary.

## 4. Mitochondrial Autophagy and PD

### 4.1. Mitochondrial Autophagy

The removal and degradation of impaired mitochondrial components is essential to mitochondrial quality control. Small, locally damaged mitochondrial components are removed when portions of the mitochondrial membrane bud off and form mitochondrial-derived vesicles (MDVs) [110]. MDVs are then transported away from their parent mitochondria and sent to lysosomes along microtubules for degradation [32]. Currently, MDV formation is believed to be regulated by microtubule-associated motor proteins 1 and 2 (Miro1/2) and DRP1: Miro1/2 induce the formation of thin mitochondrial membrane protrusions that are pulled along microtubules, while DRP1-dependent scission cleaves MDVs from the parent mitochondria [111]. Additionally, Parkin is functionally associated with MDV trafficking, mediating their transport to lysosomes and aiding in efficient degradation [112].

Canonical mitophagy is the selective autophagic degradation of entire mitochondria. It allows cells to remove damaged mitochondria, adjust mitochondrial levels to accommodate cellular demands, and maintain a steady-state turnover of mitochondria [113]. In mitophagy, targeted mitochondria are sequestered into double-membrane-bound autophagosomes. These autophagosomes are transported to lysosomes, with which they fuse, and mitochondria subsequently undergo degradation by lysosomal hydrolytic enzymes [114]. Mitophagy can be induced by various forms of stress, including oxidative stress, DNA damage, growth factor deprivation, and more [115]. Under oxidative stress, mitochondria depolarize and lose their membrane potential, which eventually triggers the start of the mitophagy pathway [116]. One of the most well-characterized mitophagy pathways is regulated by PINK1/Parkin: under physiological conditions, PINK1 is imported across the mitochondrial membrane and rapidly degraded, keeping its levels in check. However, in depolarized mitochondria, PINK1 phosphorylates ubiquitin, thus recruiting Parkin to ubiquitinate OMM proteins and trigger mitophagy [117,118,119]. Disruptions to these mitochondrial autophagy pathways can result in improper mitochondrial clearance, causing the accumulation of defective organelles and leading to neuronal damage.

### 4.2. αS Effects on Mitochondrial Autophagy in PD

Defective mitochondrial autophagy is intricately linked to PD, possibly via αS-induced alterations to MDV processes and/or to whole-mitochondria mitophagy. A recent study, for example, concluded that the overexpression of A53T αS in human neuroblastoma cells or rat primary cortical neurons induces mitophagy, causing mitochondrial dysfunction as an early event that contributes to neurodegeneration [120]. Conversely, it was reported that the normal function of αS is needed for normal mitochondrial fusion and function, with a special emphasis on the ER–mitochondria interplay [121].

A study of postmortem PD brains mechanistically demonstrated that the abnormal expression of Miro was correlated with the expression of αS and was consistent with an αS-related increase in Miro expression observed in both human neurons and a Drosophila model overexpressing αS [122]. Miro proteins typically facilitate mitochondrial motility by anchoring mitochondria to microtubules, and their removal (and the resulting lack of motility) constitutes an early step in mitochondrial clearance. Conversely, the abnormal retention of Miro may prolong active transport and inhibit mitochondrial degradation [123]. Reducing Miro levels in vivo rescued dopaminergic neuronal survival in the Drosophila while protecting their locomotion and flying ability, confirming that the phenotype was caused by the overexpression of Miro [122]. αS was found to be incorporated into the Miro complex, potentially stabilizing Miro and preventing it from being removed from the OMM, which would delay mitophagy. Additionally, αS composed of only amino acids 42 through 140 was unable to bind to Miro1, suggesting that the N-terminus of αS is required to upregulate or stabilize Miro [122]. Another study examining skin fibroblasts from PD patients revealed that over 94% of PD cell lines failed to extract Miro1 despite depolarization, further affirming the view that αS stabilizes Miro at the OMM [124]. A recent study in yeast concluded that the deletion of the yeast Miro homolog partially protects cells from the effects of A53T mutant αS [125]. Since Miro is also involved in the formation of MDVs, αS-induced Miro stabilization may affect multiple aspects of mitochondrial clearance (Figure 1c).

Studies have suggested that αS may result in the downregulation of Parkin, thus affecting several aspects of mitophagy. The treatment of a neuronal cell model with exogenous αS caused a reduction in Parkin levels, resulting in alterations to mitophagy as evidenced by the decreased ubiquitination of mitochondrial proteins and the accumulation of abnormal mitochondria [83]. The overexpression of Parkin subsequently rescued cells from these phenotypes, indicating that αS’s downregulation of Parkin interfered with mitophagy [83]. Additionally, it was proposed that exogenous αS oligomers affect the expression and activity of Parkin by causing post-translational modifications to Parkin [84]. αS may thus negatively impact several aspects of mitophagy by interfering with normal Parkin function, including Parkin’s role in MDV trafficking and Parkin’s ubiquitination of OMM proteins to initiate autophagosome formation (Figure 1c). It was recently proposed that the αS C-terminus plays a special role in mediating the PINK/Parkin interplay [106]. Moreover, another recent study highlighted the potential effect of αS on DRP1 by showing that its levels were reduced in A53T human αS transgenic mice. This was associated with two distinct phenotypes of enlarged neuronal mitochondria [126]. Others reported that αS binding to mitochondria causes SIRT3 downregulation that is accompanied by the increased phosphorylation of DRP1, which is indicative of impaired mitochondrial dynamics [127].

αS may also be involved in the trafficking and fusing of autophagosomes to lysosomes. Physiologically, the cellular actin cytoskeleton plays a significant role in trafficking autophagosomes to lysosomes in mitophagy [128]. A very recent study involving multiple model systems suggested that elevated αS levels potently suppress mitophagic flux, while non-mitochondrial autophagy is preserved. PINK1 and Parkin activation were not affected by αS excess, while the overexpression of the actin-severing protein cofilin or the inhibition of the actin-related protein 2/3 (Arp2/3) complex rescued the phenotype [129]. Others suggested that αS aggregates accelerate the translocation of cofilin-1 to mitochondria, ultimately causing oxidative damage and apoptosis [130]. In Drosophila models of synucleinopathy, αS was demonstrated to bind to spectrin, an intracellular cytoskeleton protein, promoting the reorganization of the spectrin cytoskeleton and stabilizing the actin cytoskeleton (Figure 1c) [131,132]. This stabilization resulted in mislocalization of key proteins involved in autophagosome maturation and trafficking, leading to downstream neurotoxicity [131,132]. Furthermore, alterations to the actin skeleton could disrupt cellular trafficking on a global scale, which may influence MDVs on several levels. As such, interactions between αS and the cytoskeleton may have widespread influence on mitochondrial clearance. αS may also impair mitophagy during the process of autophagosome–lysosome fusion, which is typically mediated by a SNARE complex comprising syntaxin 17 (STX17), synaptosome-associated protein 29 (SNAP29), and vesicle-associated membrane protein 8 (VAMP8) [133,134]. An investigation of αS in autophagy turnover in cultured human DA neurons revealed that αS overexpression may compromise lysosomal fusion by decreasing the abundance of SNAP29 (Figure 1c) [135]. Further investigation of postmortem LB pathology brain tissue also revealed a reduction in SNAP29 in SNpc DA neurons [135]. STX17 and VAMP8, in contrast, have not been studied well in relation to αS. Regardless, αS may interact with MDV formation and trafficking, as well as autophagosome formation, trafficking, and fusion with lysosomes, in numerous ways. A recent study suggested that both αS overexpression and knock-out in primary neurons affect the axonal transport of proteins and organelles, including mitochondria and lysosomes [136]. Since mitophagy is essential to maintaining mitochondrial health and overall cellular homeostasis, further exploration of αS’s role in mitophagy will undoubtedly offer further insight into PD pathogenesis.

## 5. Conclusions and Future Directions

αS and MQC are both key factors in PD, as indicated by environmental and genetic evidence, as well as PD models both in vitro and in vivo. The precise mechanisms linking αS with MQC are not fully understood, and neither is the chronological order in which αS problems and mitochondrial problems occur, but recent research has made significant headway, highlighting the potential effects of αS on mitochondrial biogenesis, mitochondrial dynamics, and mitochondrial autophagy. The interaction of αS with PGC-1α, PINK1/Parkin/PARIS, OPA1, and DRP1 demonstrated the ability to influence mitochondrial biogenesis and fission/fusion dynamics, but the findings will need to stand the test of time. Furthermore, αS may interact with Miro, Parkin, SNARE complexes, and/or the actin cytoskeleton to alter mitochondrial autophagy at multiple different stages. Most research on αS and MQC has looked at overexpressed αS and/or pathological αS to explore what occurs in diseased states. Examining the relationship between endogenous αS and various components of MQC may provide further insight into αS’s physiological role in MQC, thus illuminating where mitochondrial maintenance goes wrong in PD and leading to the identification of potential therapeutic targets. Additionally, while this review intentionally focused on αS effects on MQC in PD, it seems plausible that mitochondrial dysfunction conversely contributes to αS misfolding and aggregation, as mutations in certain mitochondrial genes are generally known to cause typical PD (including αS aggregation). For example, energy deficiencies within neurons may be accompanied by αS misfolding due to reduced chaperone activity, or the reduced degradation of αS due to reduced autophagy and/or proteasomal degradation [36]. Related to this, it has been proposed that mitochondria-mediated αS transfer may occur between cells, contributing to the cell-to-cell spread of αS aggregates and disease propagation [137]. Additionally, mitochondrial damage and elevated ROS may trigger neuroinflammation, affecting the overall cell cycle in the context of PD [56], and hypoxia has been discussed as a missing link between αS biology, mitochondrial dysfunction, and neurodegeneration in PD [138]. Other emerging topics include the interplay between αS, mitochondria, and fatty acid-binding proteins [139,140] or the role that αS may play in the formation, maintenance, and function of mitochondria-associated ER membranes (MAMs) [104,121,141,142,143,144]. Regardless, many lines of research support that the idea that an interplay between αS, MQC, and mitochondria in general contributes to PD, and future work in this area is critical.

## Figures and Tables

**Figure 1 biomolecules-14-01649-f001:**
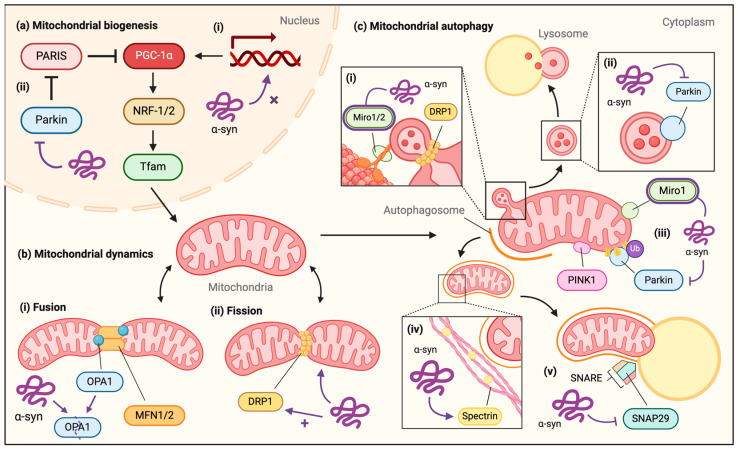
Potential pathological interactions of αS with mitochondrial quality control pathways in PD. (**a**) αS and mitochondrial biogenesis. (**i**) αS may act as a transcriptional modulator of PGC-1α under oxidative stress, binding to its promoter sequence to repress PGC-1α function. (**ii**) An excess of exogenous αS oligomers and fibrils may interfere with Parkin’s degradation of PARIS, thus increasing the PARIS-mediated transcriptional repression of PGC-1α. (**b**) αS and mitochondrial dynamics. (**i**) Pathogenic αS may increase the cleavage of OPA1 in mitochondrial fusion. (**ii**) αS-induced alterations to mitochondrial fission may be independent of or dependent upon DRP1: αS may interact directly with mitochondrial membranes or may increase the translocation of DRP1 to mitochondria. (**c**) αS and mitochondrial autophagy. (**i**) The overexpression of αS may stabilize Miro proteins, which are required for the formation of mitochondrial-derived vesicles (MDVs). (**ii**) αS may downregulate Parkin expression and activity as described above, having negative impacts on MDV trafficking. (**iii**) During autophagosome formation in mitophagy, αS may aberrantly stabilize Miro at the OMM as previously described, causing delays in mitophagy initiation. αS may also impact autophagosome formation by causing a reduction in Parkin levels, affecting the ubiquitination of mitochondrial proteins. (**iv**) By binding to spectrin, αS may excessively stabilize the actin cytoskeleton, resulting in the mislocalization of key proteins involved in autophagosome trafficking. This mislocalization may also have global effects, disrupting other forms of cellular trafficking. (**v**) Overexpressed αS may decrease SNAP29 activity, affecting the SNARE complex that mediates autophagosome–lysosome fusion during the last step of mitophagy. Figure created with BioRender. Partially adapted from Thorne and Tumbarello [32].

## Data Availability

Not applicable.

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
