# Peer review of "Alpha-Synuclein Effects on Mitochondrial Quality Control in Parkinson’s Disease"

_biomolecules, 2024, doi:10.3390/biom14121649_

Round 1
Reviewer 1 Report
Comments and Suggestions for Authors
Given the crutial role fo mitochondria and alpha-synuclein in pathogenesis of PD, in present review the authors try to summarize and discuss the relationship between alpha-synuclein and mitochondria quality control (MQC). This is one interesting topic. But there were a few big issues existing, which impact the readability of this review. Firstly, authors mainly discussed how alpha-synuclein affect MQC, but not much about how MQC impact alpha-synuclein biogeneis, aggregation, degradation. Secondly, "1. Introduction", should briefly introduce the background about PD, alpha-synuclein, MQC. However, only "1.1. Mitochondrial impairment in PD' was present, even no 1.2. Thirdly, the referenceces were not updated, and most of then were published years ago.
Author Response
Comment 1: “Firstly, authors mainly discussed how alpha-synuclein affect MQC, but not much about how MQC impact alpha-synuclein biogenesis, aggregation, degradation.”
Response 1: Thank you for pointing this out. The intended focus of our article was how aS influences MQC, rather than the bidirectional relationship between aS and MQC. We have clarified this focus in both the abstract (p. 1, para. 1) and introduction (p. 2, paras. 2, 5; p. 3, para. 1) of the article. We adjusted the title of our article from “Alpha-Synuclein and Mitochondrial Quality Control in Parkinson’s Disease” to “Alpha-Synuclein Effects on Mitochondrial Quality Control in Parkinson’s Disease” for further emphasis. Additionally, in the conclusion of our article, we elaborated on how MQC may impact aS and cited quite a few recent reviews on that topic (p. 8, para. 2).
Comment 2: “Secondly, "1. Introduction", should briefly introduce the background about PD, alpha-synuclein, MQC. However, only "1.1. Mitochondrial impairment in PD' was present, even no 1.2.”
Response 2: Thanks for this comment; we agree that the introduction could be restructured more clearly. Accordingly, we have split the introduction into 3 parts: 1.1. Parkinson’s Disease, 1.2. Mitochondrial Impairment in PD, and 1.3. Alpha-Synuclein: The Key Protein in PD. These changes, made on pp. 1-2 of the article, should provide sufficient background about the topic of the article: PD, aS, and MQC.
Comment 3: “Thirdly, the referenceces were not updated, and most of then were published years ago.”
Response 3:
We have done a Pubmed Search for “synuclein AND mitochondria” and now included in the revised manuscript all 2020-2024 publications that appeared relevant to the question how αS affects mitochondrial integrity.
Reviewer 2 Report
Comments and Suggestions for Authors
The article by Shen and Dettmer provides a synthetic but comprehensive overview of the state-of-the-art literature dealing with the involvement of mitochondria, and especially mitochondrial quality control and mitochondrial dynamics in Parkinson’s disease. The review focuses on the impact of alpha-synuclein misfolding and oligomerization on mitochondrial physiology. The authors summarize recent findings on mitochondrial biogenesis, dynamics, and autophagy in the context of PD pathology, as well as genetic and environmental evidence linking mitochondrial dysfunction and alpha-synuclein -related neurodegeneration.
Overall, the paper is clear and well-written, with no major issues that would prevent its recommendation for publication. However, I do have a couple of considerations.
First, my primary concern is the novelty of the review. Several reviews touch on mitochondrial dynamics and quality control in Parkinson’s disease, focusing on mechanisms like PINK1 and Parkin, (doi.org/10.1186/s13024-020-00367-7, https://doi.org/10.1038/s41392-023-01503-7, doi: 10.3389/fnagi.2022.885500 and others)
The review by Shen and Dettmer discusses well-characterized pathways (PINK1/Parkin, mitophagy, and mitochondrial dynamics), without covering novel aspects such as novel mechanistic links between mitochondrial dysfunction and alfa synuclein misfolding, or new therapeutic strategies targeting mitochondria.
Including insightful and original interpretation of data in the literature could differentiate this review from other recent publications on the topic.
Second, while the authors briefly mention the possibility that mitochondrial dysfunction might represent the trigger of αalfa synuclein aggregation, deepening this point could provide a more comprehensive view of the bidirectional relationship between mitochondrial dysfunction and αalfa synuclein aggregation in PD. This could also reveals additional therapeutic strategies focused on mitochondrial recovery
Author Response
Reviewer 2:
Comment 1: “First, my primary concern is the novelty of the review. Several reviews touch on mitochondrial dynamics and quality control in Parkinson’s disease, focusing on mechanisms like PINK1 and Parkin, (doi.org/10.1186/s13024-020-00367-7, https://doi.org/10.1038/s41392-023-01503-7, doi: 10.3389/fnagi.2022.885500 and others). The review by Shen and Dettmer discusses well-characterized pathways (PINK1/Parkin, mitophagy, and mitochondrial dynamics), without covering novel aspects such as novel mechanistic links between mitochondrial dysfunction and alfa synuclein misfolding, or new therapeutic strategies targeting mitochondria.”
Response 1: Thanks for this feedback, the main focus of our article was the links between aS and mitochondrial dysfunction, not mitochondria and aS dysfunction. However, to ensure we provided a fuller picture of the relationship between aS and MCQ, we briefly discussed how mitochondrial dysfunction/damage might impact aS in the conclusion (p. 8) and cited some recent reviews on that topic. We chose not to include general mitochondrial therapeutic strategies in the scope of our review. However, we have now done a comprehensive Pubmed Search for “synuclein AND mitochondria” and now included in the revised manuscript all 2020-2024 publications that appeared relevant to the question how αS affects mitochondrial integrity. This also included novel therapeutic ideas such as the delivery of cell-penetrant Parkin to neurons.
Comment 2: “Second, while the authors briefly mention the possibility that mitochondrial dysfunction might represent the trigger of αalfa synuclein aggregation, deepening this point could provide a more comprehensive view of the bidirectional relationship between mitochondrial dysfunction and αalfa synuclein aggregation in PD. This could also reveals additional therapeutic strategies focused on mitochondrial recovery.”
Response 2: Thank you for this comment. Again, the original intended focus of our article was how aS influences MQC, as opposed to the bidirectional relationships between aS and MQC. To clarify that, we added/modified several sentences in both the abstract (p. 1, para. 1) and introduction (p. 2, paras. 2, 5; p. 3, para. 1) of the article. We also elaborated briefly on how MQC may influence aS in the conclusion of the article, to emphasize the two-sided nature of their relationship and highlight recent reviews regarding that area. Additionally, we changed the title of our article from “Alpha-Synuclein and Mitochondrial Quality Control in Parkinson’s Disease” to “Alpha-Synuclein Effects on Mitochondrial Quality Control in Parkinson’s Disease” (top of p. 1) for further emphasis.